# MEGF8 is a modifier of BMP signaling in trigeminal sensory neurons

Caitlin Engelhard[1], Sarah Sarsfield[1], Janna Merte[1], Qiang Wang[1], Peng Li[2†], Hideyuki Beppu[3], Alex L Kolodkin[1], Henry M Sucov[2], David D Ginty[1]*

[1]The Solomon H Snyder Department of Neuroscience, Howard Hughes Medical Institute, Johns Hopkins University School of Medicine, Baltimore, United States; [2]Broad CIRM Center for Regenerative Medicine and Stem Cell Research, University of Southern California Keck School of Medicine, Los Angeles, United States; [3]Department of Clinical Laboratory and Molecular Pathology, Graduate School of Medicine and Pharmaceutical Science, University of Toyama, Toyama, Japan

**Abstract** Bone morphogenetic protein (BMP) signaling has emerged as an important regulator of sensory neuron development. Using a three-generation forward genetic screen in mice we have identified Megf8 as a novel modifier of BMP4 signaling in trigeminal ganglion (TG) neurons. Loss of Megf8 disrupts axon guidance in the peripheral nervous system and leads to defects in development of the limb, heart, and left-right patterning, defects that resemble those observed in *Bmp4* loss-of-function mice. *Bmp4* is expressed in a pattern that defines the permissive field for the peripheral projections of TG axons and mice lacking BMP signaling in sensory neurons exhibit TG axon defects that resemble those observed in *Megf8*−/− embryos. Furthermore, TG axon growth is robustly inhibited by BMP4 and this inhibition is dependent on Megf8. Thus, our data suggest that Megf8 is involved in mediating BMP4 signaling and guidance of developing TG axons.

*For correspondence: dginty@
jhmi.edu

†Present address: Department of Biochemistry, Stanford University, Stanford, United States

Competing interests: The authors declare that no competing interests exist.

## Introduction

During development, neurons of the peripheral nervous system (PNS) must project axons over long distances to form connections with their appropriate peripheral targets. To accomplish this task, axons of developing neurons rely on guidance provided by a variety of extracellular cues expressed in the surrounding environment and distal targets. Indeed, as developing axons extend into the periphery, they encounter a plethora of target-derived signals that they must interpret properly in order to accomplish proper axonal guidance, maturation, survival, target innervation, and synapse formation.

Bone morphogenetic protein (BMP) signaling is an important regulator of sensory neuron development. BMPs are members of the TGFβ superfamily of secreted growth factors and are required for a wide range of developmental functions including gastrulation, mesoderm formation, neural patterning, left-right asymmetry, skeletal and limb development, kidney formation, and heart development (*Zhao, 2003*). In addition to patterning, BMPs also function as axon guidance molecules. BMP7 in the spinal cord roof plate repels axons of developing commissural neurons thus orienting their growth toward the floor plate (*Augsburger et al., 1999*; *Butler and Dodd, 2003*). Moreover, BMP4 is a target-derived cue that augments development of somatosensory neurons in the trigeminal ganglia (TG) and dorsal root ganglia (DRG). BMP4 is expressed in the developing craniofacial region and signals retrogradely to coordinate differential gene expression and patterning along the dorsoventral axis of the trigeminal ganglion (*Hodge et al., 2007*). Furthermore, BMP4 appears to have a trophic effect on TG and DRG neurons as altering the level of BMP4 expression in the skin leads to changes in the peripheral innervation and number of sensory neurons found in both the TG and DRG (*Guha et al., 2004*).

**eLife digest** The peripheral nervous system relays information between the brain and spinal cord (the central nervous system) and the rest of the body. During development, neurons of the peripheral nervous system must extend processes (axons) long distances to reach the cells that they will eventually form connections with. Signaling molecules tell neuronal processes which direction to move in, and also tell them when they have reached their intended destination.

One group of molecules involved in the extension and guidance of neuronal processes are growth factors known as bone morphogenetic proteins (BMPs). These proteins contribute to a range of developmental processes, including the formation of the limbs and the skeleton, as well as various organs. They also help to establish the correct left-right patterning of the embryo, and direct the migration of sensory neurons.

Now, Engelhard et al. have used a genetic screen to identify additional signaling molecules involved in the development of the peripheral nervous system. They screened mice with a range of mutations, and found that animals with a mutant form of the gene that codes for a protein called MEGF8 closely resembled mice that lacked a member of the BMP family, BMP4. These mutants showed abnormal development of the skeleton and heart, and had six or seven digits on each limb (polydactyly).

Given the similarities between mice that lacked the gene for BMP4 and those that lacked the gene for MEGF8, Engelhard et al. explored these parallels further, and the results of a series of experiments were consistent with the two proteins being part of the same signaling cascade. In addition to identifying a novel signaling molecule that is involved in the formation of the peripheral nervous system, Engelhard et al. have provided new insights into the mechanisms by which one of the best known developmental signaling cascades is regulated.

Thus, BMP4 signaling contributes to sensory neuron maturation and target innervation; however, the mechanisms by which BMP4 acts on developing sensory neurons to promote axonal growth remain to be elucidated.

We performed a forward genetic screen in the mouse to identify novel regulators of PNS development (*Merte et al., 2010*). One of the lines that emerged from this screen has a mutation in the gene *Megf8*; loss of Megf8 function leads to defasciculation of the ophthalmic branch of the trigeminal nerve as well as defects in development of the heart, limb, skeleton, and left-right asymmetry. Given these phenotypes, we hypothesized that Megf8 interacts with the BMP4 signaling pathway. Indeed, loss of BMP signaling in trigeminal neurons leads to ophthalmic nerve defects that resemble those observed in *Megf8* loss-of-function mouse lines. Furthermore, TG axon growth is robustly inhibited by BMP4, and this inhibition is dependent on Megf8 expression. Taken together, these results show that Megf8 is a novel mediator of BMP signaling and is required for guidance of developing TG axons by target-derived BMP4.

## Results

### A forward genetic screen identifies line 687, which exhibits defasciculation of trigeminal ganglion axons

We performed ENU mutagenesis and a recessive three-generation forward genetic screen in the mouse to identify novel growth and guidance cues for developing PNS axons (*Merte et al., 2010*). One of the mutant mouse lines, Line 687, exhibits severe defasciculation of the ophthalmic branch of the trigeminal nerve. In Line 687 mutants, the two major branches of the ophthalmic nerve initially form properly as tightly bound fascicles; however, as the axons extend into the target field, they prematurely defasciculate within the face (*Figure 1A*). We mapped the genetic lesion responsible for the phenotype in Line 687 to a 2.6 Mb region of Chromosome 7 between *rs3715453* and *D7JHMI24* that contains 76 open reading frames (*Figure 1B*). Based on bioinformatic analysis of genes within this region, we sequenced the 5′UTRs and coding exons of nine genes: *Tmsb10*, *Dmrtc2*, *Zfp574*, *Zfp526*, *Gsk3a*, *Erf*, *Megf8*, *BC024561*, and *2310004L02Rik*. This analysis revealed a single base pair substitution, 5324T>C, in the gene *multiple EGF-like-domains 8* (*Megf8*) (*Figure 1C*). Megf8 is a very

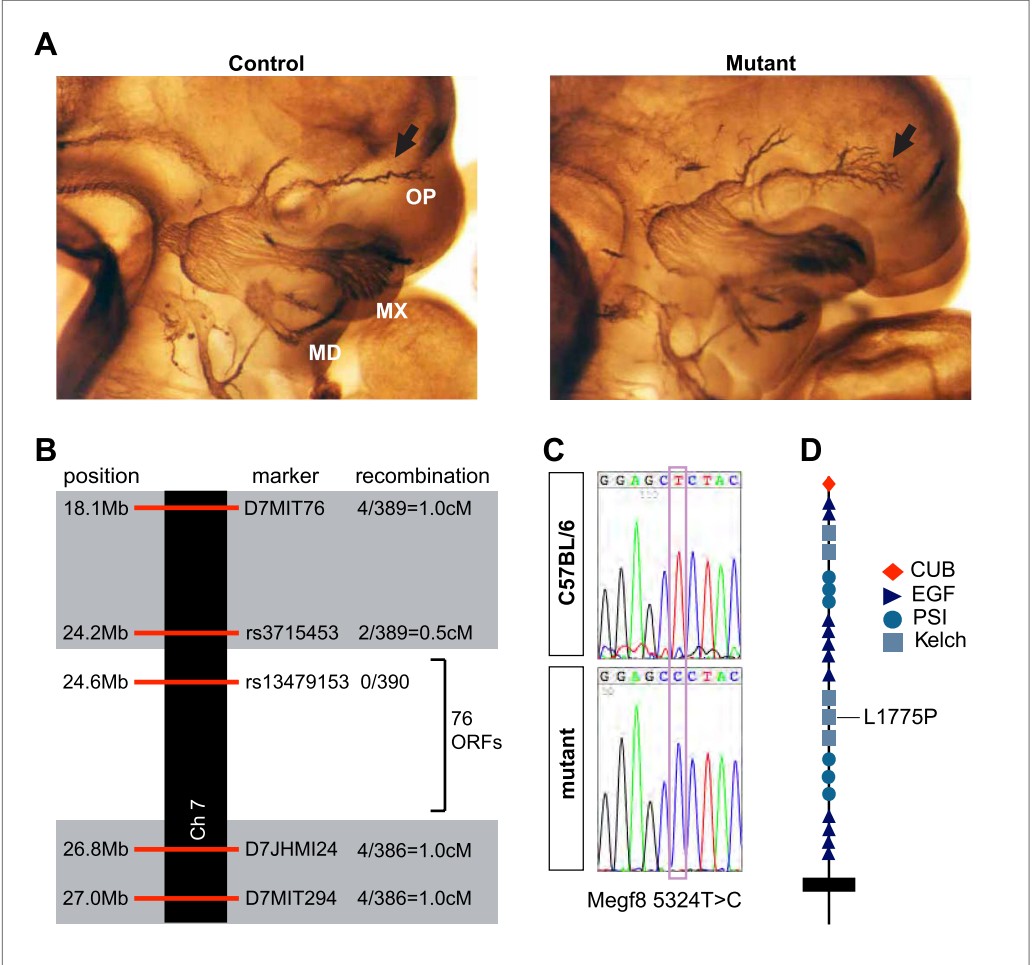

**Figure 1**. Disruption of Megf8 causes defasciculation of the TG ophthalmic nerve. (**A**) Whole-mount neurofilament staining of E11.5 control and Line 687 mutant littermates, showing the trigeminal ganglia (TG) and its three main projections: the ophthalmic (OP), maxillary (MX), and mandibular (MD) branches. (**B**) Schematic diagram of the region of Chromosome seven found to contain the Line 687 mutation, the markers used to diagnose linkage, and the frequency of recombination events observed at these markers. ORF, open reading frame. (**C**) Sequence data highlighting the mutation (*Megf8* 5324T>C) observed in Line 687 mutant DNA compared with C57BL/6 wild-type DNA. (**D**) Schematic diagram of Megf8. The Line 687 point mutation induces a single amino acid substitution L1775P in a Kelch domain of Megf8.

The following figure supplements are available for figure 1:

**Figure supplement 1**. Complementation analysis of *Megf8^{Trap}* and *Megf8^{L1775P}* alleles.

large putative transmembrane protein, and it is predicted to have a signal peptide, a CUB domain, plexin-semaphorin-integrin (PSI) domains, Kelch domains, EGF/EGF-like domains, and a C-terminal transmembrane domain. The mutation in Line 687 (*Megf8^{L1775P}*) is predicted to encode a single L>P amino acid substitution (L1775P) in the fourth Kelch domain of Megf8 (*Figure 1D*).

In order to confirm that the Line 687 TG phenotype results from the *Megf8^{L1775P}* mutation, we performed complementation analysis using a *Megf8* secretory gene-trap line obtained from the German Gene Trap Consortium (clone G037A09, *Megf8^{Gt(CD2-neo)GGTC}* or *Megf8^{Trap}*). In this line, a CD2-neomycin secretory trap vector was virally inserted between exons two and three of *Megf8*; this approach results in a null, or hypomorphic, allele of *Megf8* in which a CD2-neomycin fusion protein is brought to the cell surface instead of the endogenous protein (*Figure 1—figure supplement 1*). *Megf8^{L1775P/Trap}* embryos phenocopy the *Megf8^{L1775P/L1775P}* mutants and show defasciculation of the TG ophthalmic nerve (*Figure 1—figure supplement 1*). These results confirm that the Line 687 TG phenotype results

from the *Megf8^{L1775P}* mutation and demonstrate that Megf8 is required for proper growth and guidance of the TG ophthalmic nerve.

## *Megf8* is widely expressed during early embryonic development

We next examined the expression of *Megf8* by in situ hybridization (ISH) using wild-type embryos. Whole-mount ISH of E8.5–E10.5 embryos showed that *Megf8* is widely expressed, with strong expression in the somites, limb buds, primordial gut, developing eye, and in the pharyngeal arches (*Figure 2A–C*). *Megf8* is also highly expressed in the developing nervous system. Sensory neurons of the DRG and TG show strong expression of *Megf8* throughout embryogenesis and into the postnatal period (*Figure 2D,E*, *Figure 2—figure supplement 1*). *Megf8* is also expressed in the CNS including the developing neuroepithelium (*Figure 2F*), postnatal hippocampus, layer 4/5 of the cortex, and the olfactory bulb (*Figure 2—figure supplement 1*).

## Megf8 is required for development of the heart, limbs, skeleton, left-right asymmetry, and PNS

To gain insight into Megf8's function during development we generated a conditional knockout mouse line using the Cre/LoxP system. We targeted the final exon of *Megf8*, which encodes the transmembrane domain, intracellular C-terminus, and 3'UTR, to generate *Megf8^{Flox}* conditional mutant mice (*Figure 3—figure supplement 1*). We then crossed *Megf8^{Flox}* mice with the transgenic *CMV-Cre* line, in which cre recombinase is expressed in all cells, including germ cells (*Schwenk et al., 1995*),

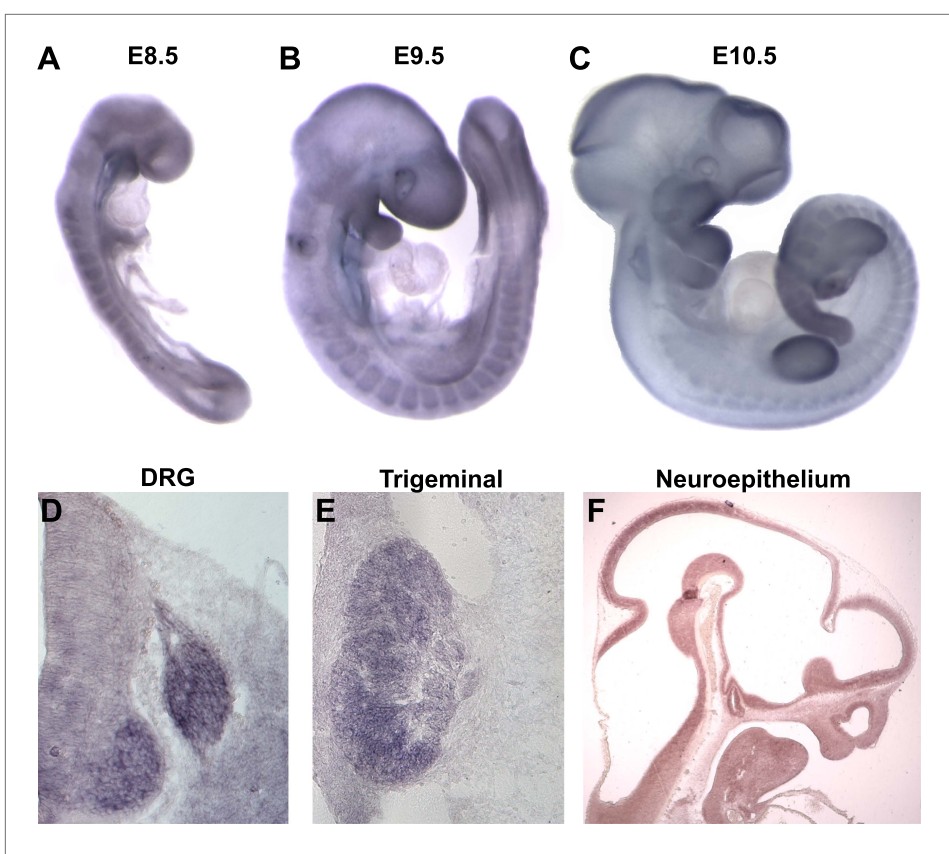

**Figure 2**. *Megf8* is expressed widely during development. (**A**–**C**) Whole-mount in situ hybridization (ISH) for *Megf8* at E8.5, E9.5, and E10.5. (**D**) ISH for *Megf8* on E11.5 transverse cryosection shows expression in the DRG. (**E**) ISH for *Megf8* on E11.5 transverse cryosection shows expression in the TG. (**F**) ISH for *Megf8* on E10.5 paraffin sagittal section shows expression in the developing neuroepithelium.
The following figure supplements are available for figure 2:

**Figure supplement 1**. *Megf8* is expressed throughout the developing nervous system.

to generate a *Megf8* null allele (*Megf8⁻*). *Megf8⁻/⁻* embryos phenocopy *Megf8^L1775P/L1775P* embryos and exhibit dramatic defasciculation and undergrowth of the ophthalmic branch of the trigeminal nerve (**Figures 3A and 4H**). In both the *Megf8* null and *Megf8^L1775P/L1775P* lines the maxillary and mandibular branches of the TG are largely unaffected. The maxillary branch shows no defasciculation phenotype although it is slightly undergrown, while the mandibular branch shows no abnormality (**Figure 4H**, legend).

Consistent with its widespread expression during embryogenesis, we observed that loss of *Megf8* leads to severe developmental abnormalities in several organ systems. Both *Megf8⁻/⁻* and *Megf8^L1775P/L1775P* embryos display polydactyly in both the forelimbs and hindlimbs; this pre-axial polydactyly is completely penetrant, and embryos show six to seven digits per limb as well as duplication of bones in the hand (**Figure 3B**, **Figure 3—figure supplement 2**). They also exhibit skeletal abnormalities including delayed ossification of the rib cage, a wider and shorter rib cage, and a split sternum (**Figure 3C**, **Figure 3—figure supplement 2**). Furthermore, loss of *Megf8* leads to a disruption of left-right patterning. Approximately one third of *Megf8⁻/⁻* embryos demonstrate a complete left-right inversion of heart looping (**Figure 3D**) or embryonic turning (**Figure 3E**). Taken together with prior results, these findings indicate that Megf8 is required for development of left-right asymmetry.

In addition to its role in development of the limb, skeleton, and left-right asymmetry, Megf8 is also required for normal heart development. In addition to the inversion of heart looping observed in some *Megf8⁻/⁻* and *Megf8^L1775P/L1775P* embryos, loss of Megf8 function leads to a complete, or nearly complete, absence of the mitral and tricuspid valves, swollen atria, transposition of the outflow tract, pulmonary stenosis, and atrial and ventricular septal defects (**Figure 3—figure supplement 3**). These phenotypes were seen in all embryos at E14.5 or later, and they lead to severe peripheral edema (**Figure 3F**) and embryonic lethality by age E16.5. Analysis at earlier stages revealed apparently normal endocardial cushion mesenchymal cells (**Figure 3—figure supplement 3**), which implies that the terminal pathologies result from improper remodeling events after E12.5. Conditional mutation of the *Megf8^Flox* allele using the *Mesp1-Cre* line (**Saga et al., 1999**), which is active in all mesoderm of the heart, did not cause any heart phenotypes (data not shown), implying that early heart and embryo laterality defects might be responsible for the constellation of later heart defects. These findings are consistent with a second mouse line harboring a *Megf8* point mutation (*Megf8^C193R*) that was identified in an independent forward genetic screen and which was shown to have polydactyly and defects in left-right patterning and heart development (**Zhang et al., 2009**).

Because *Megf8* is expressed strongly throughout the nervous system, we next investigated the function of Megf8 in the developing PNS, in addition to its role in the TG. In *Megf8⁻/⁻* embryos, spinal nerves are shorter at E11.5 than in wild type (**Figure 4A,D**), and by E13.5 the radial and ulnar nerves have not fully extended into the limbs, appearing shorter and less branched (**Figure 4B,E**). At E13.5 the limbs appear immature in *Megf8⁻/⁻* embryos relative to wild type littermates, which may contribute to the radial and ulnar nerve phenotypes. *Megf8⁻/⁻* embryos also display defects in the developing vagus and glossopharyngeal nerves. At E11.5 these nerves are slightly defasciculated and several wayward axons branch away from the main fiber bundle (**Figure 4C,F**).

Overall, these findings demonstrate that Megf8 is required for proper axonal extension in several areas of the PNS, including the TG, DRG, and vagus/glossopharyngeal nerves (**Table 1**). Given the broad expression of *Megf8* in a wide range of tissues during development, we next asked whether these defects, and in particular the TG defect, are due to a cell autonomous function of Megf8 in sensory neurons. To address this question, we crossed the conditional *Megf8^Flox* line with the transgenic *Wnt1-Cre* line (**Danielian et al., 1998**) in which cre recombinase is expressed in neural crest-derived cells, including sensory neurons. *Wnt1-Cre*-mediated conditional deletion of *Megf8* leads to defasciculation of the ophthalmic branch of the trigeminal nerve, phenocopying *Megf8⁻/⁻* and *Megf8^L1775P/L1775P* mutant embryos (**Figure 4G**). Although all *Wnt1-Cre;Megf8^Flox/Flox* embryos showed defasciculation of the ophthalmic branch, the severity of the phenotype was more variable than in *Megf8⁻/⁻* and *Megf8^L1775P/L1775P* embryos (**Figure 4H**), and in a few embryos the ophthalmic branch defasciculated completely around the eye and failed to project into the periphery (data not shown). These findings suggest that Megf8 mediates guidance of developing TG axons in a cell autonomous manner. *Wnt1-Cre;Megf8^Flox/Flox* embryos do not exhibit a defect in radial/ulnar nerve development, however, suggesting that Megf8's role in spinal nerve axon guidance is non-cell autonomous (**Figure 4—figure supplement 1**). Of note, *Wnt1-Cre;Megf8^Flox/Flox* embryos do not exhibit the limb immaturity seen

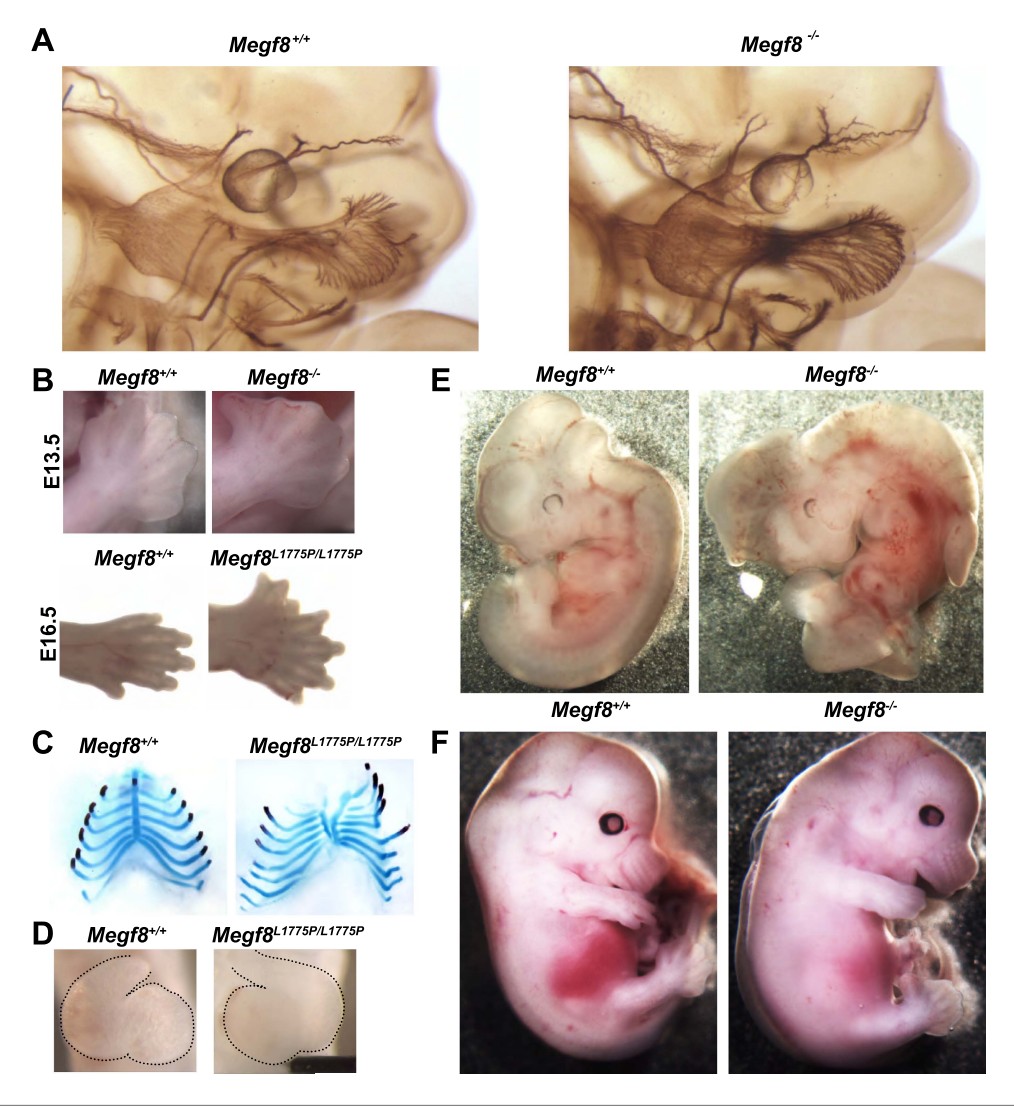

**Figure 3**. Megf8 is required for development of the trigeminal ganglia, limb, skeleton, heart, and left–right asymmetry. (**A**) Whole-mount neurofilament staining of E11.5 *Megf8*+/+ and *Megf8*−/− littermates. The *Megf8*−/− null mutant phenocopies the point mutant *Megf8*L1775P/L1775P. (**B**) Whole-mount images of *Megf8*+/+ and *Megf8*−/− hindlimbs at E13.5 (top) and *Megf8*+/+ and *Megf8*L1775P/L1775P forelimbs at E16.5 (bottom). (**C**) Alcian blue and alizarin red staining of E16.5 embryonic ribs/sternum. *Megf8*L1775P/L1775P mutants have a split sternum and delayed ossification of the rib cage. (**D**) Whole-mount images of the heart of freshly fixed E10.5 embryos with dissected pericardial cavity. *Megf8*L1775P/L1775P have complete left–right inversion of heart looping. Heart is outlined with dotted lines. (**E**) Whole-mount images of E11.5 *Megf8*+/+ and *Megf8*−/− littermates, showing reversal of embryonic turning and exencephaly in the *Megf8*−/−. (**F**) Whole-mount images of *Megf8*+/+ and *Megf8*−/− littermates at E13.5, showing severe edema in the *Megf8*−/−.

The following figure supplements are available for figure 3:

**Figure supplement 1**. Generation of a conditional knock-out mouse line (*Megf8*Flox).

**Figure supplement 2**. *Megf8*L1775P/L1775P mutants show defects in limb and skeletal development.

**Figure supplement 3**. Heart development in *Megf8*L1775P/L1775P mutant embryos.

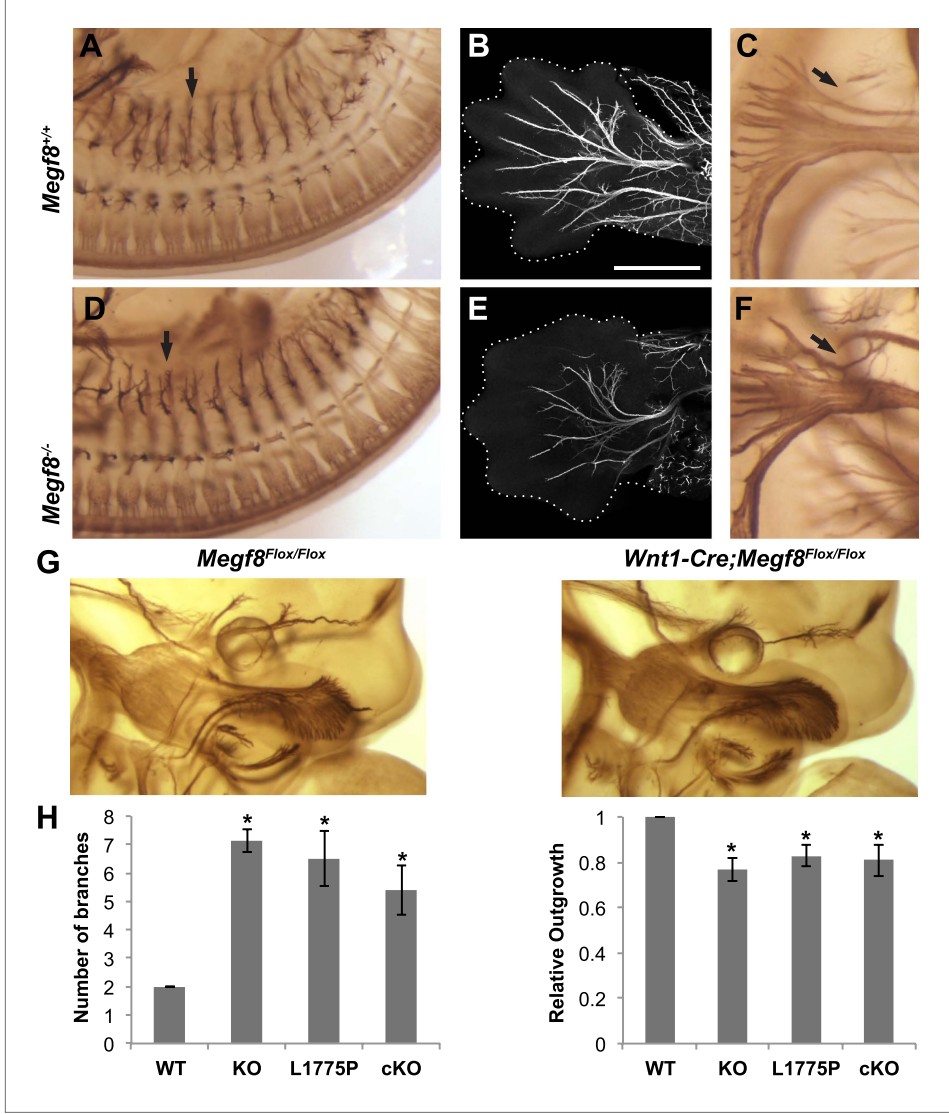

**Figure 4**. Megf8 is required for development of the PNS. (**A** and **D**) Whole-mount neurofilament staining of E11.5 *Megf8+/+* and *Megf8–/–* littermates showing the DRG spinal nerves, which are undergrown in the *Megf8–/–* (arrow). (**B** and **E**) Whole-mount peripherin staining of forelimbs from E13.5 *Megf8+/+* and *Megf8–/–* littermates. The radial and ulnar nerves are undergrown in the *Megf8–/–* embryo. Limbs are outlined with dotted lines. Scale bar (**B** and **E**) represents 500 μm. (**C** and **F**) Whole-mount neurofilament staining of E11.5 *Megf8+/+* and *Megf8–/–* littermates. The vagus/glossopharyngeal nerves are defasciculated in the *Megf8–/–* (arrow). (**G**) Whole-mount neurofilament staining of E11.5 *Megf8Flox/Flox* and *Wnt1-Cre; Megf8Flox/Flox* littermates. (**H**) Quantification of ophthalmic branch phenotype for *Megf8–/–* (KO), *Megf8L1775P/L1775P* (L1775P), and *Wnt1-Cre; Megf8Flox/Flox* (cKO) compared to *Megf8+/+* (WT). Left: the number of branches at the nasociliary branch point was significantly greater for KO, L1775P, and cKO embryos. Right: the ophthalmic branch was undergrown in KO, L1775P, and cKO embryos. Four to seven embryos were analyzed per genotype. Error bars represent mean ± s.e.m. *p<0.05, one-way ANOVA. The relative outgrowth was also measured for the maxillary and mandibular branches of the TG for *Megf8–/–* embryos (not shown). The maxillary branch was slightly undergrown compared to *Megf8+/+* (relative outgrowth 0.9, p<0.05) while the mandibular branch was unaffected (relative outgrowth 0.99, p=0.7). To assess defasciculation in the maxillary branch, the relative maxillary area was calculated and no difference was observed between *Megf8+/+* and *Megf8–/–* (relative area 0.94, p=0.2).

The following figure supplements are available for figure 4:

**Figure supplement 1**. Conditional deletion of *Megf8* from DRG neurons does not disrupt formation of the radial/ulnar nerves.

**Table 1.** Summary of *Megf8−/−* phenotypes and implicated BMPs

| Phenotype | *Megf8-/-* | BMP implicated | |
|---|---|---|---|
| Trigeminal nerve (V1) defasciculation | 100% (10/10) | n.d. | |
| Trigeminal patterning | n.d. | BMP4 | (*Hodge et al., 2007*) |
| Polydactyly | 100% (9/9) | BMP4, BMP7 | (*Dudley et al., 1995*; *Dunn et al., 1997*) |
| Reversed heart looping | 33% (7/21) | BMP4 | (*Fujiwara et al., 2002*) |
| Reversed embryonic turning (E11.5) | 33% (5/15) | BMP4 | (*Fujiwara et al., 2002*) |
| Edema (E13.5+) | 100% (6/6) | | |
| DRG spinal nerves undergrown (E11.5) | 100% (10/10) | BMP4 | (*Guha et al., 2004*) |
| Radial/ulnar nerves undergrown (E13.5) | 100% (2/2) | BMP4 | (*Guha et al., 2004*) |
| Vagus defasciculation | 90% (9/10) | n.d. | |
| Exencephaly | 36% (16/45) | | |
| Disrupted *BMP4* expression around V1 | 100% (4/4) | | |

Phenotypes, stages of observation and penetrances of *Megf8−/−* mutants are listed. V1 refers to the ophthalmic branch of the trigeminal nerve. *Bmp* loss-of-function lines known to display similar phenotypes are noted.

in *Megf8−/−* embryos. Thus, one possibility is that the radial/ulnar nerve phenotype observed in *Megf8−/−* embryos is secondary to an overall delay in limb development. An alternative explanation is that Megf8 may be required in spinal motor neurons, and that a loss of motor neuron axon guidance in *Megf8−/−* embryos causes a subsequent disruption in DRG sensory axon guidance; this indirect effect of Megf8 on DRG sensory neuron axon guidance would not be evident in *Wnt1-Cre;Megf8Flox/Flox* embryos.

### *Megf8−/−* phenotypes resemble *Bmp* loss-of-function lines

Our analysis demonstrates that Megf8 is required for development of the limb, skeleton, heart, left-right asymmetry, and PNS. The spectrum of phenotypes seen in *Megf8−/−* embryos is strikingly similar to that observed when BMP signaling is disrupted (*Table 1*). BMPs are members of the TGF-β superfamily of extracellular ligands and have been implicated in a wide range of developmental functions. We noted that loss of BMP4, and to a lesser extent BMP2 and BMP7, results in similar defects as those observed in the *Megf8* mutants. *Bmp4* and *Bmp7* loss-of-function mouse lines exhibit pre-axial polydactyly (*Dudley et al., 1995*; *Luo et al., 1995*; *Dunn et al., 1997*; *Selever et al., 2004*). *Bmp4* null embryos also display disrupted left-right patterning and loss of normal rightward heart looping (*Fujiwara et al., 2002*). Loss of BMP4, or its receptor BMPR2, results in a wide range of heart defects, including atrial septal defects, ventricular septal defects, atrioventricular septal defects, and abnormal positioning or septation of the outflow tract (*Jiao et al., 2003*; *Liu et al., 2004*; *Beppu et al., 2009*); several of these cardiac phenotypes are exacerbated by a concomitant loss of BMP2 or BMP7 (*Rivera-Feliciano and Tabin, 2006*; *Goldman et al., 2009*). Furthermore, BMP4 signaling is required for dorsoventral patterning of the TG (*Hodge et al., 2007*), and modifying BMP4 expression in the skin leads to changes in the peripheral innervation and survival of sensory neurons (*Guha et al., 2004*). Taken together, our findings demonstrate that loss of either BMP4 or Megf8 leads to a similar spectrum of defects in the limb, left-right asymmetry, and heart, and that both BMP4 and Megf8 are required for normal development of the TG. Based on these functional similarities, we hypothesized that Megf8 interacts with, or is a component of, the BMP4 signaling pathway, and that this interaction is required for axon guidance and peripheral target innervation by TG sensory neurons.

### *Bmp4* is expressed in peripheral targets in a manner that defines a permissive field for trigeminal nerve axonal projections

To address the hypothesis that Megf8 and BMP4 cooperate to regulate growth and guidance of TG axons, we began by assessing the spatial relationship between *Bmp4* expression and developing TG axons. To do so, we utilized a *Bmp4lacZ* mouse line in which a *lacZ* reporter cassette was inserted into the *Bmp4* locus (*Lawson et al., 1999*). Using whole-mount β-galactosidase and neurofilament

co-staining, we were able to examine the expression pattern of *Bmp4* relative to developing TG axons. At E10.5, *Bmp4* is expressed strongly in the eye and also in the dorsal portion of the face above the eye (*Figure 5A*). These two areas of BMP4 expression flank a narrow corridor through which the developing ophthalmic nerve projects; this expression pattern suggests that BMP4 may act as

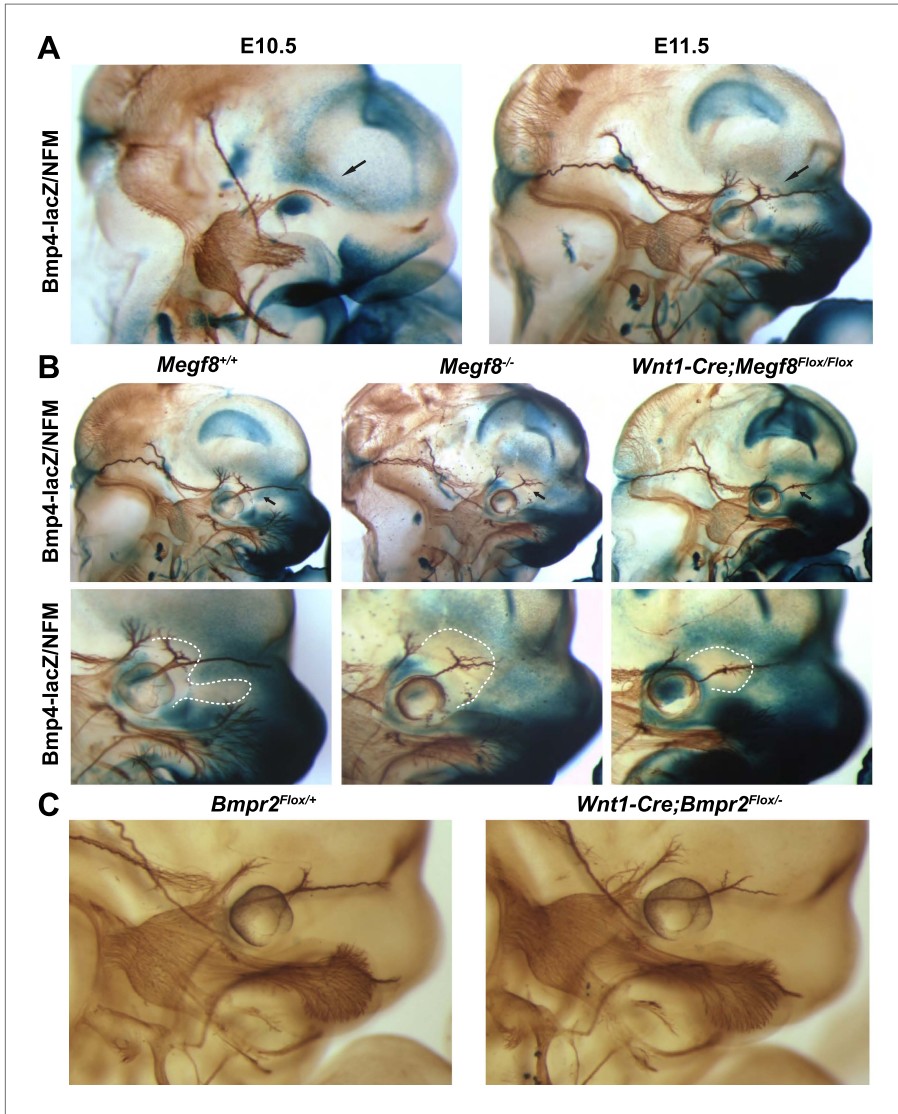

**Figure 5**. BMP signaling is required for proper extension of the TG ophthalmic nerve. (**A**) Whole-mount β-galactosidase and neurofilament (NFM) co-staining on *Bmp4^lacZ/+* embryos at E10.5 and E11.5, showing the relationship between *Bmp4* expression and the developing TG nerve. (**B**) Whole-mount β-galactosidase and neurofilament co-staining on E11.5 embryos: *Bmp4^lacZ/+;Megf8^+/+* (left) and *Bmp4^lacZ/+;Megf8^−/−* (center) littermates, as well as *Bmp4^lacZ/+;Wnt1-Cre/Megf8^Flox/Flox* (right). *Bmp4^lacZ* expression is lost at the location where defasciculation of the TG ophthalmic nerve occurs in *Megf8^−/−* and *Wnt1-Cre;Megf8^Flox/Flox* embryos. The top row shows the whole head with all three TG branches. The bottom row is an enlargement of the top row showing the ophthalmic branch. The area of perturbed *Bmp4* expression is outlined. The disruption of *Bmp4^lacZ* expression was fully penetrant and observed in all *Megf8^−/−* (n = 4) and *Wnt1-Cre;Megf8^Flox/Flox* (n = 3) embryos assessed. (**C**) Whole-mount neurofilament staining of E11.5 *Bmpr2^Flox/+* and *Wnt1-Cre;Bmpr2^Flox/−* littermates.

The following figure supplements are available for figure 5:

**Figure supplement 1**. *Bmp4* expression in *Megf8^−/−* embryo compared to wild-type littermate.

**Figure supplement 2**. *Bmpr2* and *Megf8* are expressed throughout the developing TG.

a repulsive guidance cue for developing TG axons. One day later, at E11.5, the TG axons have extended into their peripheral targets and have bifurcated into frontal and nasociliary nerves. At this stage, *Bmp4* expression is striking; as the ophthalmic branch extends beyond the eye, the nasociliary nerve branches in a region devoid of *Bmp4* expression and then immediately projects nasally as a tightly bundled fascicle through an area of strong *Bmp4* expression while the axons of the frontal branch extend rostrally. Remarkably, the pattern of *Bmp4* expression is disrupted in *Megf8*−/− embryos. In the absence of Megf8, *Bmp4* expression is lost at the point of defasciculation of the ophthalmic branch into the frontal and nasociliary branches (*Figure 5B*; *Table 1*) suggesting that axonal innervation of the target region is necessary for proper expression of *Bmp4* and that the absence of this BMP signaling leads to defasciculation of the nasociliary branch of the ophthalmic nerve. An alternative explanation is that Megf8 is required in cells of the craniofacial target tissues to regulate *Bmp4* expression. ISH shows that *Megf8* is expressed in craniofacial tissues as well as the TG, although *Megf8* expression is far more robust in neurons of the TG (*Figure 2*). Interestingly, in *Megf8*−/− embryos *Bmp4* expression is disrupted exclusively in the area around the ophthalmic branch but is intact throughout the rest of the embryo (*Figure 5—figure supplement 1*).

## Mice lacking BMP signaling in TG sensory neurons exhibit axon guidance defects similar to those of *Megf8*−/− embryos

To directly test the idea that BMP4 is required for the growth and guidance of TG axons, we next sought to assess development of the trigeminal nerve in the absence of BMP4 signaling. Because *Bmp4* null embryos die too early in development for our analysis, we used mice lacking the BMP receptor, BMPR2, in cells of the neural crest lineage. We utilized mice harboring one mutant BMP receptor conditional allele (*Bmpr2Flox*) and one null allele (*Bmpr2−*) (*Beppu et al., 2005*) crossed to *Wnt1-Cre* mice to eliminate the receptor from TG sensory neurons. BMPR2, one of several BMP receptors, is preferentially activated by BMP ligands and not by other TGF-β superfamily members (*Miyazono et al., 2010*). *Bmpr2* is robustly expressed in all three lobes of the developing TG, as is *Megf8* (*Figure 5—figure supplement 2*). Interestingly, the ophthalmic branch of the trigeminal nerve in E11.5 *Wnt1-Cre;Bmpr2Flox/−* embryos is stunted and branches prematurely in a manner reminiscent of the phenotype seen in *Megf8* loss-of-function lines (*Figure 5C*). Furthermore, the maxillary and mandibular branches of the trigeminal nerve are unaffected in the *Wnt1-Cre;Bmpr2Flox/−* embryos, as in *Megf8* null lines. These results demonstrate that BMP signaling is required specifically for development of the ophthalmic branch of the trigeminal nerve, and they suggest that both Megf8 and BMPR2 are necessary to mediate this signaling.

## BMP4 inhibits TG axon outgrowth, and this inhibition is dependent on *Megf8* expression

We next asked whether BMP4 and Megf8 functionally interact to mediate growth and guidance of TG axons. To test this idea, we used an in vitro TG explant assay in which TG explants were cultured in a collagen gel and exposed to increasing amounts of bath applied BMP4. Remarkably, BMP4 robustly inhibits the outgrowth of wild-type TG axons, and maximal doses inhibit axon outgrowth by more than 50% (*Figure 6A,C*). We then asked whether loss of *Megf8* would alter the response of TG axons to BMP4 by co-culturing explants from *Megf8*−/− embryos alongside littermate controls. *Megf8*−/− explants exhibit significantly greater axon outgrowth in the presence of BMP4, but not under control culture conditions (*Figure 6B,D*). These results demonstrate that *Megf8*−/− axons are less sensitive to the inhibitory effects of BMP4 and that the response of TG neurons to BMP4 is partially dependent on *Megf8* expression. Together with our in vivo findings, these results support a model in which Megf8 functions within TG sensory axons, allowing them to respond to intermediate target-derived BMP4 and thus guide these axons to their appropriate peripheral target regions in the face.

## Discussion

Forward genetics provides a powerful approach to identify new axon guidance cues and also novel components of signaling pathways already known to regulate axon growth and guidance. In this study, we used an unbiased three-generation forward genetic screen to identify the gene *Megf8* as a novel regulator of PNS development. Our findings strongly suggest that Megf8 regulates the response of target-derived BMP4 on TG axons during development and that functional Megf8 signaling is necessary for proper TG axon growth and target innervation.

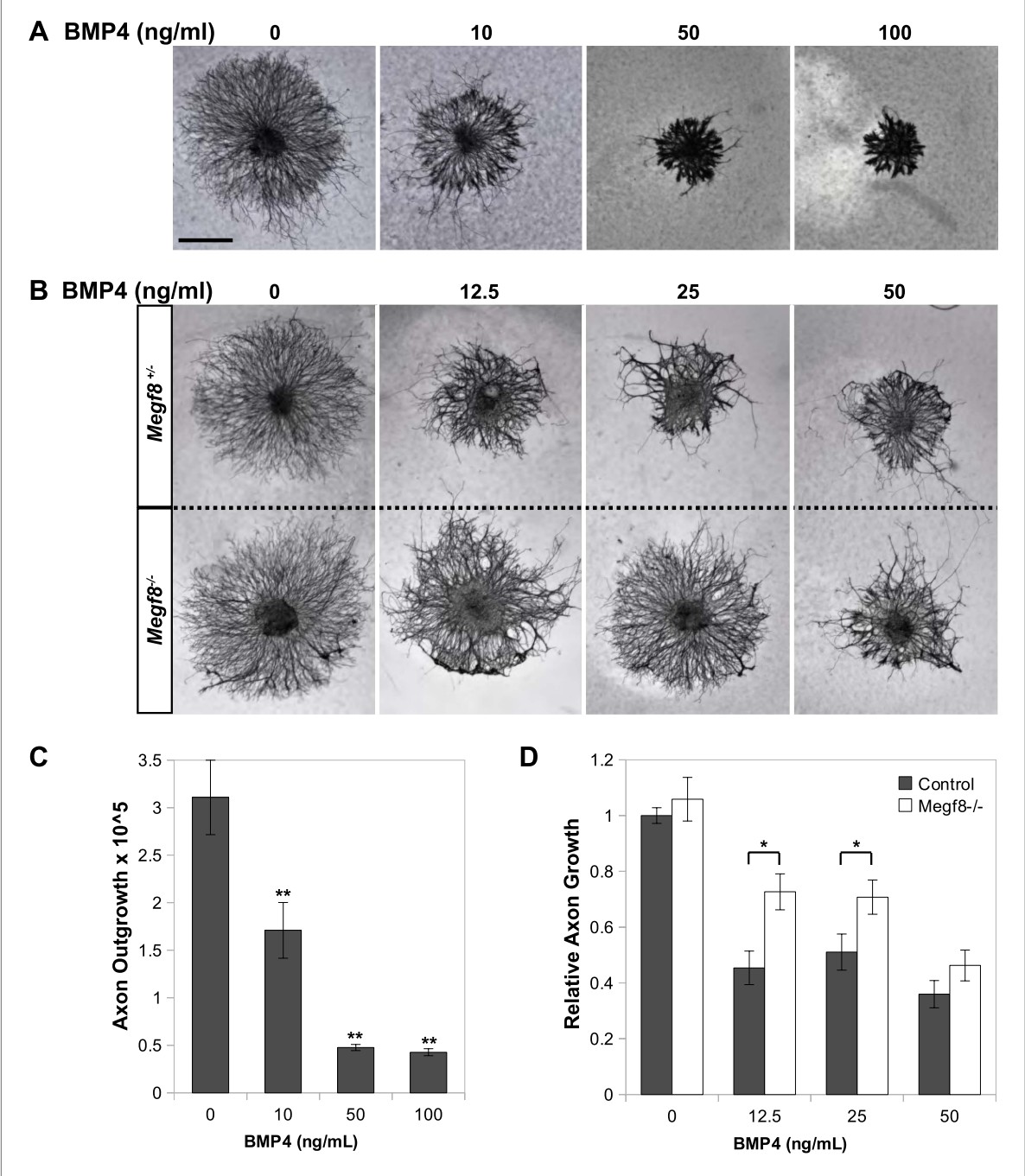

**Figure 6**. Megf8 mediates the inhibition of TG axon growth by BMP4. (**A**) E12.5 wild-type TG explants cultured in 0–100 ng/ml BMP4. (**B**) E12.5 TG explants from *Megf8+/–* and *Megf8–/–* littermates cultured side by side in 0–50 ng/ml BMP4. (**C**) Quantification of (**A**): axonal outgrowth of wild-type TG explants in the presence of 0–100 ng/ml BMP4. Increasing doses of BMP4 caused robust inhibition of axon outgrowth (**p<0.05, Student's *t* test). Three to four explants were quantified for each concentration of BMP4. Error bars represent mean ± SEM. (**D**) Quantification of (**B**): relative axon outgrowth of TG explants from control (*Megf8+/+* or *Megf8+/–*) and *Megf8–/–* littermates in the presence of 0–50 ng/ml BMP4. BMP4 inhibition of axon outgrowth is partially lost in *Megf8–/–* explants (*p<0.05, Student's *t* test). The experiment was repeated three times, using three to four explants per BMP4 concentration in each experiment; each bar represents results from at least 10 explants. Error bars represent mean ± SEM. Scale bar represents 20 μm.

## Megf8 regulates development of the limb, skeleton, heart, left-right asymmetry, and PNS

Analysis of *Megf8^{L1775P/L1775P}* and *Megf8^{−/−}* embryos demonstrates that Megf8 functions broadly during embryogenesis and is required for the development of several organ systems. Loss of Megf8 function leads to pre-axial polydactyly, skeletal defects, disruption of left-right patterning, and severe heart defects, all of which together result in embryonic lethality (present study, and *Zhang et al., 2009*). Interestingly, mutations in *Megf8* were recently identified in a small subset of children with Carpenter syndrome, which is an autosomal-recessive multiple-congenital-malformation disorder. Mutations were identified in several domains of *Megf8*, including the Kelch domain adjacent to the Kelch domain harboring the L1775P point mutation, and resulted in thoracic skeletal defects, limb defects such as pre-axial polydactyly, congenital heart defects, and profound lateralization defects including transposition of the great arteries, dextrocardia, and complete situs inversus (*Twigg et al., 2012*). These phenotypes mirror those seen in our *Megf8* loss-of-function lines and reinforce the vital role for *Megf8* throughout embryogenesis.

Megf8 is also required for development of the peripheral nervous system, including the growth and guidance of the TG, spinal, and vagus/glossopharyngeal nerves. *Megf8^{−/−}* embryos exhibit profound defasciculation of the ophthalmic branch of the TG, undergrown and underbranched spinal nerves, and defasciculation of the vagus/glossopharyngeal nerves. *Megf8* is strongly expressed in sensory neurons throughout embryogenesis, and conditional deletion of *Megf8* in neural crest-derived cells, including sensory neurons (*Wnt1-Cre;Megf8^{Flox/Flox}*), leads to a defect in the TG ophthalmic nerve that phenocopies the *Megf8* null and *Megf8^{L1775P/L1775P}* lines. These results strongly suggest that Megf8 functions cell autonomously in TG neurons to regulate axon guidance and innervation of peripheral targets.

## Megf8 is a novel modifier of BMP4 signaling in trigeminal ganglion neurons

The phenotypes observed in *Megf8^{L1775P}* and *Megf8^−* mouse lines show a striking resemblance to those observed in mice lacking BMP4 signaling (*Table 1*). Like the *Megf8^{−/−}* and *Megf8^{L1775P/L1775P}* embryos, loss of BMP4 function leads to pre-axial polydactyly (*Dunn et al., 1997*), loss of left-right patterning including reversed heart looping (*Fujiwara et al., 2002*), and severe heart defects (*Jiao et al., 2003*; *Liu et al., 2004*). BMP4 signaling has recently been implicated in PNS development, where it is required by neurons in the TG for specifying positional identity (*Hodge et al., 2007*) and in both the TG and DRG for peripheral target innervation and survival (*Guha et al., 2004*).

Given these phenotypic similarities, we hypothesized that Megf8 is a component of the BMP4 signaling pathway and specifically regulates the response of TG neurons to target-derived BMP4. We assessed *Bmp4* expression during E10.5–E11.5, which is the period when TG axons extend into their peripheral target fields and found that *Bmp4* expression appears to define the permissible field for ophthalmic nerve target innervation suggesting that BMP4 may function as a repulsive guidance cue. Consistent with these findings, conditional deletion of the BMP receptor, *Bmpr2*, in sensory neurons using *Wnt1-Cre;Bmpr2^{Flox/−}* mice results in defasciculation of the ophthalmic nerve and phenocopies the *Megf8* mutant lines. Thus, axons of TG ophthalmic sensory neurons require both *Megf8* and BMP signaling to correctly navigate towards their target field. Remarkably, in each of the *Megf8* and *Bmpr2* loss-of-function lines, the TG maxillary and mandibular branches are largely unaffected despite widespread expression of Megf8 and BMPR2 within the TG ganglion, which demonstrates that Megf8 and BMPR2 function selectively in neurons of the TG ophthalmic branch to mediate axon guidance. We suggest that Megf8 is a mediator or a key component of the BMPR2 signaling pathways that control growth and guidance of ophthalmic nerve axons.

The expression pattern of *Bmp4* relative to the developing ophthalmic nerve suggests that BMP4 acts as an inhibitory cue to prevent developing TG axons from innervating inappropriate targets. This hypothesized role for BMP4 is supported by our in vitro findings, in which BMP4 treatment of TG explant cultures resulted in robust inhibition of axon outgrowth. This inhibition was partially lost in *Megf8^{−/−}* explants, indicating that TG neurons require *Megf8* expression to mediate their maximal response to the inhibitory effects of BMP4. These findings suggest that Megf8 is necessary for BMP4 signaling in sensory neurons and that this signaling provides an important inhibitory cue to guide developing TG axons. In addition, our finding that *Bmp4* expression is disrupted in *Megf8^{−/−}* embryos at the site of defasciculation of the TG ophthalmic branch suggests a possible non-cell autonomous role for Megf8 in regulating *Bmp4* expression, wherein *BMP4* expression is dependent on signaling from incoming

TG axons, which is disrupted by defasciculation in the $Megf8^{-/-}$ embryo. Alternatively, BMP4 may be directly regulated by Megf8 expressed in cells of the craniofacial region. Given that $Bmp4$ expression is preserved in all other areas of the $Megf8^{-/-}$ embryos as well as our in vitro findings that show Megf8 is required in TG neurons to mediate the inhibitory effects of exogenous BMP4, we think it is unlikely that Megf8 predominantly regulates BMP4 signaling through modifying $BMP4$ expression. Future studies will address the Megf8 mechanism of action in developing neurons and, in particular, whether it is a mediator of expression or axonal localization of BMPR2, BMP4 binding to BMPR2, or BMP4–BMPR2 signaling in axons.

Taken together, our findings support a model in which Megf8 is a mediator of BMP4 signaling in TG axons. $Megf8$ is expressed in TG sensory neurons where it acts cell autonomously to promote signaling in response to target-derived BMP4. Megf8-mediated BMP4 signaling, which is inhibitory for TG axons, likely enables these axons to remain fasciculated en route to their appropriate targets in the face. Disruption of this signaling, either through loss of Megf8 or other components of the BMP4 pathway, leads to a corresponding loss of inhibition and premature defasciculation of the nerve. Thus, BMP4 signaling is required for TG axon guidance and Megf8 is a novel component of BMP4 signaling in TG sensory neurons. Given the numerous phenotypic similarities between $Megf8$ and $Bmp$ loss-of-function mouse lines, including defects in the limb, heart, and left-right patterning, we propose that Megf8 mediates signaling by BMP4 and, perhaps, other TGFβ family members throughout the embryo during development.

## Materials and methods

### Whole-mount neurofilament staining

Isolated embryos were fixed overnight in 4% paraformaldehyde/PBS at 4°C. All subsequent washes and incubations were performed at room temperature on an orbital shaker. After fixation, embryos were washed in PBS (three 10 min washes) and dehydrated as follows: 50% methanol/PBS (1 hr), 80% methanol/PBS (2 hr), and 100% methanol (overnight). Endogenous peroxidase activity was quenched by incubating the embryos overnight in a solution of 3% hydrogen peroxide, 70% methanol, and 20% DMSO. The embryos were next washed in five 45 min incubations in TNT (10 mM Tris-base, 154 mM NaCl, 0.1% Triton X-100) and then incubated for 3 days in primary antibody solution, which consisted of 2H3 anti-neurofilament ascites antibody (3.8 mg/ml stock, Developmental Studies Hybridoma Bank, University of Iowa, Iowa City, IA) diluted 1:5000 in TNT with 0.02% sodium azide, 4% milk, 5% DMSO, and 2% sheep serum (Millipore, Billerica, MA). Following primary antibody, embryos were washed in TNT (five time, 45 min each), and then incubated for 2 days in secondary sheep anti-mouse IgG HRP-conjugated antibody (Jackson ImmunoResearch Laboratories, West Grove, PA) diluted 1:250 in TNT with 0.02% sodium azide, 4% BSA, 5% DMSO, and 2% sheep serum. The embryos were washed in TBS for four 45 min incubations followed by one overnight incubation. The following morning, the horseradish peroxidase reactions were developed with diaminobenzidine, washed in five incubations of TBS for 5 min each, and dehydrated through methanol as done previously (50%, 80%, 100% methanol). To visualize the staining, embryos were cleared in 1:2 benzyl alcohol/benzyl benzoate.

### Quantification of trigeminal nerve phenotype

In order to assess the defasciculation and outgrowth of the three trigeminal nerve branches, images of representative embryos stained with the whole-mount neurofilament assay were analyzed using ImageJ software (*Abramoff et al., 2004*). The ophthalmic branch defasciculation was assessed by counting the number of branches that emerged from the branch point of the nasociliary nerve distal to the eye. The outgrowth of all three trigeminal nerve branches (ophthalmic, maxillary, and mandibular) was quantified by measuring the distance the branch travelled into the peripheral tissue after exiting the TG. This distance was then divided by the outgrowth measured in a littermate control in order to calculate the relative outgrowth of the nerve. In order to assess for possible defasciculation in the maxillary nerve, the area of the maxillary branch was measured beginning at the point of exit from the TG and extending to its innervation of peripheral tissues. This area was then divided by the area measured in a littermate control to calculate the relative area of the maxillary branch.

### RNA in situ hybridization

Digoxigenin (DIG)-labeled cRNA probes were used for in situ hybridization. To generate probes directed against $Megf8$, 1.1 kb and 2.5 kb ApaI fragments from MGC clone 5369271 (Thermo Scientific,

Pittsburgh, PA) were cloned into pBK-CMV (Agilent, Santa Clara, CA). A probe for *Bmpr2* was amplified using gene-specific PCR primers (Allen Institute for Brain Science, Seattle, WA) and cDNA template prepared from P4 mouse brain. The resulting fragment was cloned into pCRII-TOPO (Life Technologies, Carlsbad, CA). Embryos were fixed in 4% paraformaldehyde/PBS overnight at 4°C. For whole-mount preparations, embryos were then dehydrated into methanol and in situ hybridization was performed as described in *Parr et al. (1993)* with modifications (*Knecht et al., 1995*). For sectioned tissue, after fixation embryos were washed in PBS, cryoprotected in 30% sucrose, embedded and frozen in Tissue-Tek OCT (Sakura Finetek USA, Inc., Torrance, CA), and serially sectioned at 20 μm. In situ hybridization was then performed on sectioned tissue as described previously (*Schaeren-Wiemers and Gerfin-Moser, 1993*). To assess the developing neuroepithelium, in situ hybridization was performed on formalin-fixed, paraffin-embedded tissue.

## Skeletal analysis

Cartilage and bone were stained with alcian blue 8GX (Sigma-Aldrich, St. Louis, MO) and alizarin red S (Sigma-Aldrich) as previously described (*McLeod, 1980*).

## Whole-mount peripherin staining on limb

E13.5 embryos were eviscerated and fixed in 4% paraformaldehyde/PBS overnight at 4°C. The forelimbs and hindlimbs were then separated and immunostained with rabbit anti-peripherin (Millipore, Billerica, MA, 1:2000) as described in *Wickramasinghe et al. (2008)*.

## Whole-mount β-galactosidase and neurofilament double staining

Isolated embryos were fixed in 4% paraformaldehyde/PBS at 4°C for 30 min. After fixation, the embryos were washed three times with X-gal rinse solution (100 mM sodium phosphate pH7.3, 2 mM MgCl$_2$, 0.01% sodium deoxycholate, and 0.02% NP40) and then stained overnight at room temperature in X-gal staining solution (5 mM potassium ferricyanide, 5 mM potassium ferrocyanide, and 1 mg/ml X-gal in X-gal rinse solution). They were then post-fixed in 4% paraformaldehyde/PBS at 4°C overnight to improve the stability of the β-galactosidase stain. Following β-galactosidase staining and post-fixation, the embryos were immunostained according to the whole-mount neurofilament protocol outlined above.

## Trigeminal explant assay and axon growth analysis

Trigeminal ganglia (TG) were dissected from E12.5 embryos. The dorsal half of each TG was isolated in order to select for neurons that give rise to the ophthalmic branch. Explants were then cultured in collagen droplets in Neurobasal media (Life Technologies, Carlsbad, CA) supplemented with B27, 20 ng/ml NGF, and 0–100 ng/ml BMP4 (R&D Systems, Minneapolis, MN). After 40 hr in culture, the explants were fixed in 4% paraformaldehyde overnight at 4°C. They were then immunostained for neurofilament as follows. After fixation, explants were washed in PBS (three 15 min washes), permeabilized in 0.25% Triton X-100/PBS for 1 hr at room temperature, incubated overnight at room temperature with 2H3 anti-neurofilament ascites antibody (Developmental Studies Hybridoma Bank, University of Iowa, Iowa City, IA) diluted 1:5000 in blocking buffer (5% sheep serum, 5% BSA, and 0.25% Triton X-100 in PBS), washed in 0.25% Triton X-100/PBS (five 15 min washes), incubated overnight at room temperature with secondary sheep anti-mouse IgG HRP-conjugated antibody (Jackson ImmunoResearch Laboratories, West Grove, PA) diluted 1:250 in blocking buffer, washed in 0.25% Triton X-100/PBS (five 15 min washes), and finally developed with diaminobenzidine to visualize staining. Images of the explants were quantified using ImageJ software (*Abramoff et al., 2004*). In order to determine the axon outgrowth, we quantified the explant's total area and then subtracted the small portion of this area that comprised the cell bodies; the remaining area represented the total axon outgrowth for a given explant. The relative axon growth was calculated by dividing an explant's total axon outgrowth by the average outgrowth for control explants (control genotype and 0 ng/ml BMP4 treatment).

## Mice and genotyping

The *Bmp4^lacZ* (*Lawson et al., 1999*), BMP receptor (*Bmpr2*) conditional knockout (*Beppu et al., 2005*), *Wnt1-Cre* (*Danielian et al., 1998*), and *Mesp1-Cre* (*Saga et al., 1999*) mouse lines were generated and genotyped as previously described. The Line 687 point mutant (*Megf8^L1775P*) encodes a restriction enzyme-sensitive polymorphism and was genotyped by amplification of an approximately 600 bp fragment using the primers 5′-GGCAAGGGATGTGGTCATTAAG and 5′-CTTTTCTCCAAACCAGGCATGGAAA, followed by SacI digestion. The *Megf8^Trap* line was genotyped by PCR using the primers 5′-GACCACCAGCCTGAGTGCAAAA and

5′-TCGAGCACAGCTGCGCAAGGAA. The *Megf8^Flox* allele was genotyped using the primers 5′-AAGCCTGGATGCAAGGGCAGAT and 5′-CATAGGGCCACCATCAGCTCAT. The *Megf8^−* allele was genotyped using the primers 5′-AAGCCTGGATGCAAGGGCAGAT and 5′-CATAGGGCCACCATCAGCTCAT.

## Acknowledgements

We thank members of the Ginty laboratory for discussions throughout the course of this study and for reading and commenting on the manuscript. We thank Brigid Hogan (Duke University) for providing the *Bmp4^LacZ* mice.

## Additional information

### Funding

| Funder | Grant reference number | Author |
| --- | --- | --- |
| Howard Hughes Medical Institute | | Alex L Kolodkin, David D Ginty |
| National Institutes of Health | NS34804 | David D Ginty |
| National Institutes of Health | HL078891 | Henry M Sucov |
| National Institutes of Health | HL070123 | Henry M Sucov |
| National Institutes of Health | MH59199 | Alex L Kolodkin, David D Ginty |

The funders had no role in study design, data collection and interpretation, or the decision to submit the work for publication.

### Author contributions

CE, Conception and design, Acquisition of data, Analysis and interpretation of data, Drafting or revising the article; SS, QW, ALK, HMS, Acquisition of data, Analysis and interpretation of data, Drafting or revising the article; JM, Conception and design, Acquisition of data, Analysis and interpretation of data; PL, HB, Acquisition of data, Analysis and interpretation of data; DDG, Senior author, designed the project and oversaw all aspects of the science and the preparation of the manuscript, Conception and design, Analysis and interpretation of data, Drafting or revising the article

### Ethics

Animal experimentation: This study was done in accordance with the recommendations in the Guide for the Care and Use of Laboratory Animals of the National Institutes of Health. All of the animals were handled according to approved institutional animal care and use committee (IACUC) protocols of the Johns Hopkins University School of Medicine. The protocols were approved by the Animal Care and Use Committee of the Johns Hopkins University School of Medicine (Protocol Numbers: MO11M10 and MO09M412).

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
