## [Decision Letter]

Thank you for sending your work entitled “MEGF8 is a modifier of BMP signaling in trigeminal sensory neurons” for consideration at *eLife*. Your article has been favorably evaluated by a Senior editor, a Reviewing editor, and 2 reviewers.

The following individuals responsible for the peer review of your submission have agreed to reveal their identity: Marianne Bronner (Reviewing editor); Samantha Butler and Bill Snider (peer reviewers).

The Reviewing editor and the two reviewers discussed their comments before we reached this decision, and the Reviewing editor has assembled the following comments to help you prepare a revised submission.

This is a straightforward and nicely done study. By screening for peripheral nerve defects after ENU mutagenesis, the authors have identified a mutation in the *Megf8* gene. Mutation of the human homologue (Carpenter syndrome) is associated with dramatic developmental abnormalities. Here, the authors describe the cloning of *Megf8* as well as the careful and thorough characterization of its loss-of-function. Megf8 appears to be a novel receptor with critical roles in axon guidance, the establishment of left-right asymmetry and limb, skeletal, and heart development. Because the mouse and human phenotypes bear similarity to loss of BMP4 signaling, the authors performed a comprehensive analysis of the requirement for Megf8 in development in comparison to the requirement for *Bmp4*.

As a result of their findings, the authors propose an important mechanistic link between Megf8 and BMP signaling: that Megf8 is required to interpret BMP4 as a repellent. *Bmp4* expression surrounds the ophthalmic branch of the trigeminal ganglion (TG) and apparently keeps the nerve fasciculated. In the absence of either Megf8 or BmprII, this branch is severely defasciculated. Moreover, loss of Megf8 results in the loss of *Bmp4* expression flanking the ophthalmic branch, in the region where the most profound defasciculation defects are observed. Taken together, these studies identify a novel putative regulator of BMP signaling and raise important questions about the non-autonomous control of *Bmp4* expression by Megf8.

Major comments:

1) This paper provides a mechanistic explanation for the defasciculation defect observed in *Megf8* mutants, but does not explain the foreshortening of many nerves, including their focus in this paper: the ophthalmic branch (Figure4H)? If removing the response to a repellent, wouldn't one predict that the axons might grow longer, as seen in Figure 6?

2) The phenotype observed in Figure 5 is very interesting, suggesting that Megf8 is required to stabilize or maintain *Bmp4* expression. Is this phenotype also fully penetrant (add to Table 1?)? Could the authors provide some quantification of *Megf8* mutants to more clearly assess the extent to which *Bmp4* activity staining is lost in the region surrounding the ophthalmic nerve compared to controls? Importantly, the authors barely discuss this very striking result in the Discussion! Doesn't it suggest that the loss of Megf8 has a non-autonomous negative effect on *Bmp4* expression? If so, how mechanistically do they think a membrane bound receptor is suppressing *Bmp4* expression?

3) Is the interaction between *Megf8* and BMP4 the likely mechanistic explanation for the other phenotypes? In which case, why is *Bmp4* expression not lost in other affected organs (Figure 5)?

4) The authors are careful to assay only the response of the ophthalmic branch of the TG nerve to BMP4 stimulation in vitro. Did they try a similar in vitro assay with the maxillary and mandibular branches? It would be interesting to know whether these branches are impervious to BMP4 in vitro as suggested by the results in vivo.

5) Similarly, presumably other classes of neurons are affected, as there are also abnormalities of spinal roots although the spinal root abnormalities were not analyzed in detail.

6) Although the link to BMP signaling is intriguing, the current evidence is rather indirect and based primarily on their one in vitro experiment. A simple way to provide strong evidence for a direct effect on BMP signaling in vivo would be to assess Phospho-Smad staining levels in mutant versus wild type trigeminal ganglia. The authors should attempt this experiment.

7) In the Discussion, it is critical that the authors add further comment on how they think Megf8 regulates BMP4 signaling.

---

## [Author Response]

*1) This paper provides a mechanistic explanation for the defasciculation defect observed in* Megf8 *mutants, but does not explain the foreshortening of many nerves, including their focus in this paper: the ophthalmic branch (*Figure 4*)? If removing the response to a repellent, wouldn't one predict that the axons might grow longer, as seen in*
Figure 6?

There are two possibilities for the foreshortening observed in the ophthalmic branch of the trigeminal nerve. One possibility is that the foreshortening is secondary to the defasciculation. In other words, individual ophthalmic axons grow to the same length (or longer) in *Megf8* mutants, but because the axons aberrantly extend into surrounding tissues the overall path of the nerve appears shorter. The other possibility is that in addition to its role in mediating BMP4 inhibitory cues, Megf8 also facilitates the growth of ophthalmic axons into the periphery and thus these axons are undergrown in *Megf8* mutants. This second possibility is consistent with the phenotype observed in the spinal nerves, which are undergrown in *Megf8* null embryos when compared to controls (Figure 4). In addition, the maxillary branch of the trigeminal nerve is slightly undergrown in *Megf8* mutants compared to control (Figure 4 legend). Taken together, these results suggest a role for Megf8 in promoting axon growth of TG and DRG sensory neurons. However, our data do not determine whether this occurs through modification of BMP4 signaling.

*2) The phenotype observed in*
Figure 5
*is very interesting, suggesting that Megf8 is required to stabilize or maintain Bmp4 expression. Is this phenotype also fully penetrant (add to*
Table 1*?)? Could the authors provide some quantification of* Megf8 *mutants to more clearly assess the extent to which* Bmp4 *activity staining is lost in the region surrounding the ophthalmic nerve compared to controls? Importantly, the authors barely discuss this very striking result in the Discussion! Doesn't it suggest that the loss of Megf8 has a non-autonomous negative effect on* Bmp4 *expression? If so, how mechanistically do they think a membrane bound receptor is suppressing* Bmp4 *expression*?

The disruption of *Bmp4-lacZ* expression around the TG ophthalmic branch is fully penetrant in *Megf8*^*-/-*^ embryos, and this point is now made in the Table in the revised paper. Interestingly, *Bmp4-lacZ* expression is normal in all other parts of the embryo. This can be appreciated in Figure 5—figure supplement 1, which shows the *Bmp4-lacZ* expression pattern throughout *Megf8*^*-/-*^ and *Megf8*^*+/+*^ embryos and includes images of the whole embryo, DRG, and limb.

We have provided commentary on the changes in the *Bmp4-lacZ* expression pattern in the *Megf8*^*-/-*^ embryo and the implications of these changes in the revised manuscript. Our findings suggest that the loss of BMP4 expression at the site of defasciculation of the TG ophthalmic branch could result from two possible mechanisms. Either BMP4 expression is dependent on signaling from incoming TG axons (which is disrupted by defasciculation in the *Megf8*^*-/-*^ embryo) or Megf8 expressed in the craniofacial target region regulates BMP4 expression. This is an interesting point, as we know from our ISH results that *Megf8* is expressed in TG neurons as well as in the developing craniofacial area. Because neural crest cells contribute to this developing craniofacial area, the *Wnt1-Cre*; *Megf8*^*f/f*^ embryo unfortunately does not allow us to distinguish between these two possibilities.

*3) Is the interaction between* Megf8 *and BMP4 the likely mechanistic explanation for the other phenotypes? In which case, why is* Bmp4 *expression not lost in other affected organs (*Figure 5*)*?

Indeed, given the phenotypic similarities between *Megf8* mutant and *BMP4* loss of function lines, we hypothesize that the functional interaction between Megf8 and BMP4 is the mechanistic explanation for the other phenotypes observed in *Megf8* mutants, including polydactyly, disruption of left-right patterning, and heart defects. As we have shown, *BMP4* expression in *Megf8* null embryos is only disrupted around the TG ophthalmic branch; it is intact in all other areas of the embryo including the DRG, heart, and limbs. These findings suggest that Megf8 modifies BMP4 signaling by acting on the BMP4 ligand/receptor complex or downstream components of the signaling pathway, not through modifying expression of *BMP4* itself. Indeed, this is consistent with our in vitro findings (Figure 6), in which we found that in the presence of exogenous BMP4 ligand, *Megf8* null axons respond poorly to BMP4 as an inhibitory cue. If Megf8’s role in BMP4 signaling is primarily to modify *BMP4* expression, one would expect that *Megf8* null axons would still be inhibited by exogenous BMP4 ligand, unlike our results in Figure 6. In fact, we were surprised to find that BMP4 expression is disrupted around the TG ophthalmic nerve. We suggest that this finding (Figure 5) in conjunction with our in vitro analysis (Figure 6) indicates that Megf8 modifies BMP4 signaling in TG ophthalmic axons in two ways: 1) Megf8 expressed in the axon is required to mediate the BMP4 inhibitory cue and 2) loss of Meg8 in the axon, loss of Megf8 in the surrounding craniofacial tissue, or loss of normal TG axon extension (secondary to loss of Megf8) disrupts BMP4 expression. This is discussed in the revised manuscript.

*4) The authors are careful to assay only the response of the ophthalmic branch of the TG nerve to BMP4 stimulation* in vitro*. Did they try a similar* in vitro *assay with the maxillary and mandibular branches? It would be interesting to know whether these branches are impervious to BMP4* in vitro *as suggested by the results* in vivo.

Our initial explant experiments included the entire ganglion and we did see robust inhibition of axon outgrowth by BMP4 (as well as loss of this inhibition in the *Megf8*^*-/-*^ explants). However, we decided to focus on the dorsal half of the TG, which contains the ophthalmic lobe and part of the maxillary lobe, so as to enrich for ophthalmic neurons. Given the large amount of maxillary neurons present (roughly 50% of the explant), our results suggest that BMP4 can inhibit maxillary axon outgrowth, although it is unclear if BMP4 inhibits maxillary neurons to the same extent as ophthalmic neurons. We did not attempt a separate analysis of the maxillary and mandibular branches.

*5) Similarly, presumably other classes of neurons are affected, as there are also abnormalities of spinal roots although the spinal root abnormalities were not analyzed in detail*.

*Megf8* null embryos show undergrowth of DRG spinal nerves as well as defasciculation of the vagus and glossopharyngeal nerves. In addition, Megf8 is broadly expressed throughout the developing neuroepithelium (Figure 2), which suggests that it may play a role in different neuronal populations in the CNS and PNS. However, we have not yet asked whether BMP signaling is disrupted in other classes of neurons. In future experiments it will be important to assess the role of Megf8 in other axon guidance events controlled by BMP signaling, including the ventral axonal trajectory of neurons whose cell bodies reside within the dorsal spinal cord.

*6) Although the link to BMP signaling is intriguing, the current evidence is rather indirect and based primarily on their one* in vitro *experiment. A simple way to provide strong evidence for a direct effect on BMP signaling* in vivo *would be to assess Phospho-Smad staining levels in mutant versus wild type trigeminal ganglia. The authors should attempt this experiment*.

Phospho-SMAD immunohistochemistry on wild type and *Megf8* null TG was attempted but, unfortunately, we encountered a widespread pattern of very strong staining in virtually all cell types and tissues in the preparations, and very high background. Thus, we were unable to ascertain whether phospho-SMAD expression is intact or diminished in *Megf8* null embryos (which would imply that Megf8 modifies BMP4 signaling via a non-SMAD dependent mechanism), or if we could not observe a difference that was present because of issues of specificity or because this assay would not allow us to detect subtle changes in phospho-SMAD levels. We are therefore not confident that P-SMAD staining provides a specific and reliable quantitative assay for comparing *Megf8*^*-/-*^ TG to wild type littermates and have not pursued this direction.

*7) In the Discussion, it is critical that the authors add further comment on how they think Megf8 regulates BMP4 signaling*.

We have included discussion of mechanisms by which Megf8 regulates BMP signaling in the revised manuscript.